# Hot Deformation Behavior and Microstructure Evolution of Cu–Ni–Co–Si Alloys

**DOI:** 10.3390/ma13092042

**Published:** 2020-04-27

**Authors:** Feng Liu, Jimiao Ma, Lijun Peng, Guojie Huang, Wenjing Zhang, Haofeng Xie, Xujun Mi

**Affiliations:** 1State Key laboratory of Nonferrous Metals and Processes, GRIMAT Group Co., Ltd., Beijing 100088, China; liuf@cn-shine.com (F.L.); penglijun198677@163.com (L.P.); huangguojie@grinm.com (G.H.); zhangwenjing@grinm.com (W.Z.); xiehaofeng@grinm.com (H.X.); 2GRIMAT Engineering Institute Co., Ltd., No. 11 Xingkedong Str., Huairou District, Beijing 101417, China; 3General Research Institute for Nonferrous Metals, Beijing 100088, China; 4Ning Bo XingYe ShengTai Group Co., Ltd., Ningbo 315336, China; mjm@cn-shine.com

**Keywords:** Cu–Ni–Co–Si alloy, hot compression deformation, constitutive equation, dynamic recrystallization, microstructure

## Abstract

The Cu-1.7Ni-1.4Co-0.65Si (wt%) alloy is hot compressed by a Gleeble-1500D machine under a temperature range of 760 to 970 °C and a strain rate range of 0.01 to 10 s^−1^. The flow stress increases with the extension of strain rate and decreases with the rising of deformation temperature. The dynamic recrystallization behavior happens during the hot compression deformation process. The hot deformation activation energy of the alloy can be calculated as 468.5 kJ/mol, and the high temperature deformation constitutive equation is confirmed. The hot processing map of the alloy is established on the basis of hot deformation behavior and hot working characteristics. With the optimal thermal deformation conditions of 940 to 970 °C and 0.01 to 10 s^−1^, the fine equiaxed grain and no holes are found in the matrix, which can provide significant guidance for hot deformation processing technology of Cu–Ni–Co–Si alloy.

## 1. Introduction

Cu–Ni–Si alloy is an ideal material for integrated circuit lead frames, connectors, elastic components and power conversion, which has high tensile strength, high softening temperature, and excellent electrical and thermal conductivity. The C70250 alloy is a typical Cu–Ni–Si alloy with tensile strength of 600–800 MPa and electrical conductivity of 35%–45% IACS [1,2,3,4,5]. The addition of the Co element to Cu–Ni–Si alloy can form a dispersive and high heat-resistance stability (Ni, Co)_2_Si precipitation phase in the matrix, which can significantly improve the strength and high temperature softening resistance without sacrificing the conductivity of the alloy [6,7]. At present, the large-scale Cu–Ni–Si alloy ingot is prepared by a semi-continuous casting process, and the strip is obtained by hot rolling and cold rolling, as well as heat treatment. Hence, the effect of hot deformation on the microstructure and mechanical properties of copper alloy was studied in previous work [8,9,10,11]. Lei [8] has studied the thermal compression deformation behavior of the Cu-6.0Ni-1.0Si-0.5Al-0.15Mg-0.1Cr alloy under a temperature range of 700–970 °C and a strain rate range of 0.001–1 s^−1^, established the stress–strain constitutive equation and acquired the reasonable hot processing deformation parameters: 850–875 °C and 0.001–0.01 s^−1^. According to the trinary phase diagram of Cu–Ni–Si, Ni_2_Si intermetallic is precipitated at temperature 880 °C and the addition of Co to this system also results in the formation of the (Ni,Co)_2_Si phase at a temperature of 1050 °C [12]. Since the phase transition temperature and stress of (Ni, Co)_2_Si phases are higher than those of the Ni_2_Si phase, the deformation behavior, microstructure and microstructure evolution of the Cu–Ni–Co–Si alloy during hot processing may be different from those of the Cu–Ni–Si alloy, which have a great effect on the formability and comprehensive performance of the alloy.

Therefore, the high-temperature deformation behavior of the Cu-1.7 wt% Ni-1.4 wt% Co-0.65 wt% Si alloy (hereinafter referred to as Cu-1.7Ni-1.4Co-0.65Si) was studied under the conditions of a deformation temperature of 760–970 °C and a strain rate of 0.01–10 s^−1^. Moreover, the constitutive relationship of hot deformation and thermal processing maps were established to reveal the microstructure evolution and deformation mechanism in the process of hot deformation, and to provide guidance for the establishment of a hot working process for the Cu–Ni–Co–Si alloy.

## 2. Materials and Methods

The raw materials were pure Cu (99.95 wt%), pure Ni (99.99 wt%), pure Co (99.95 wt%) and pure Si (99.90 wt%). The ingot was prepared by using an intermediate-frequency induction melting furnace with the dimensions of 640 mm length, 210 mm width, and 6000 mm height. The melting temperature was about 1300 °C and the casting temperature was about 1250 °C.

The high temperature compression test was conducted on the Gleeble1500D material thermal simulation test machine (DSI company, St. Paul, MN, America). The samples were taken from alloy ingots with a diameter of 10 mm and a length of 15 mm. The hot deformation treatments were 760 °C, 790 °C, 820 °C, 850 °C, 880 °C, 910 °C, 940 °C and 970 °C, and the strain rates were 0.01, 0.1, 1 and 10 s^−1^, respectively. The samples were heated with a rate of 20 °C/s and held for 3 min. Before the experiment, lubricant (75% graphite + 20% oil + 5% trimethylbenzene nitrate) was applied to both ends of the cylinder sample, and tantalum pieces were affixed to the end of the hydraulic shaft of the equipment to reduce the influence of friction in the deformation process. The deformed specimens were quickly cooled by water to room temperature to obtain the deformed structure. The microstructure was observed with a LV150 Optical Microscope (OM, Nikon company, shanghai, China). The samples for OM observation were mechanically polished and corroded in a solution of 5 g FeCl_3_ + 10 mL HCl + 90 mL H_2_O. The samples for transmission electron microscope (TEM, JEOL Ltd., Tokyo, Japan) observations were first cut to a thickness of around 500 µm and then ground to 60 µm, after which these samples were punched into a disc of 3 mm in diameter followed by electro-polishing using a mixed solution of CH_3_OH:HNO_3_ = 4:1 under a temperature of −35–30 °C. The microstructures of different states were observed under a JEM 2100 LaB6 transmission electron microscope (TEM, JEOL Ltd., Tokyo, Japan).

## 3. Results and Discussion

### 3.1. The True Stress–True Strain Curve

Figure 1 shows the true stress–true strain curves of Cu-1.7Ni-1.4Co-0.65Si alloy at different strain rates and temperatures when the true strain rate is set as 0.8. As seen from Figure 1, the true stress raises as the strain rate increases at the same deformation temperature. However, the stress decreases as the deformation temperature increases with a constant strain rate. The true stress increases linearly with the true strain in the stage of elastic deformation with a strain rate range of 0.01 to 1 s^−1^ at various temperatures. When the stress exceeds the yield strength of those alloys, plastic deformation is observed and the stress continues to enhance as the true strain rises. The softening effect derived from the appearance of dynamic recovery and recrystallization is less than the hardening effect derived from the increase in dislocation density during the deformation process. Compared to the Cu–Ni–Si alloy [13], the addition of Co to the Cu–Ni–Si alloy can restrain dynamic recovery and recrystallization during the hot deformation. The flow stress of the Cu–Ni–Co–Si alloy is much higher than that of the Cu–Ni–Si alloy when the temperature range is 760 to 790 °C and the strain rate is 0.1 to 1 s^−1^. The hot deformed 760 °C shows smaller grains compared to the reference Cu–Ni–Si alloy [13]. This can contribute to improving the strength after hot deformation according to the Hall–Petch relationship.

When the strain rate is 10 s^−1^, the true stress–true strain curve shows an obvious serrated feature, as shown in Figure 1d. This fluctuation is mainly affected by the alternations of softening and hardening during the deformation process. In the process of hot deformation of the Cu–Ni–Co–Si alloy, the presence of precipitated particles provides a significant pinning effect on dislocation motion. With the strain rate increasing, many dislocations are pinned around the precipitated particles, which produces a large stress concentration and enhances the flow stress of the alloy. When the strength effect caused by the behavior of precipitation is less than the softening effect caused by the behaviors of dynamic recovery and recrystallization in the process of deformation, the flow stress trends to decrease.

### 3.2. Constitutive Model of Flow Stress

As a basic parameter to characterize the deformation of alloys, flow stress can be used to determine the load capacity and energy consumption during hot deformed processes. The hot process of alloys is also controlled by the thermal activation process [14]. The constitutive relation of material deformation at high temperatures can be described by two typical equations, that is, the relationship between temperature, strain rate and stress. The relationship between deformation temperature, strain rate and flow stress is usually described by the Arrhenius-type hyperbolic sine constitutive equation, which contains deformation activation energy Q and deformation temperature T. This model can be estimated as [15,16]:(1)$$Z=\dot{\varepsilon }exp(\frac{Q}{\mathrm{RT}})$$

The relationship between deformation parameters can be shown as Equation (2) for low stress and Equation (3) for high stress, respectively:(2)$$\alpha \sigma <0.8,\dot{\varepsilon }=A_{1}\sigma^{n1}exp(\frac{-Q}{\mathrm{RT}})$$
(3)$$\alpha \sigma >1.2,\dot{\varepsilon }=A_{2}exp\left(\beta \sigma \right)exp(\frac{-Q}{\mathrm{RT}})$$

For all stress levels, the hyperbolic sine Equation (4) can be used to express:(4)$$\dot{\varepsilon }=A{\left[sinh\left(\alpha \sigma \right)\right]}^{n}exp(\frac{-Q}{\mathrm{RT}})$$
where Z is the Zener–Hollomon parameter; σ is the flow stress, MPa; $\dot{\varepsilon }$ is the strain rate, s^−1^; A1, A2, A, α, n_1_ and β are material constants; R is the gas constant, 8.314 J/mol·K; Q is the activation energy, kJ/mol; T is the thermodynamic temperature, K; α = β/n_1_.

Combining Equations (1) and (3) can obtain:(5)$$Z=\dot{\varepsilon }exp(\frac{-Q}{\mathrm{RT}})=A{\left[sinh\left(\alpha \sigma \right)\right]}^{n}$$

Taking the logarithm of both sides of Equations (2) and (3) can obtain:(6)$$ln\dot{\varepsilon }=lnA_{1}+n_{1}ln\sigma -Q/RT$$
(7)$$ln\dot{\varepsilon }=lnA_{2}+\beta \sigma -Q/RT$$

Figure 2 shows the relationship curve between ln$\dot{\varepsilon }$-lnσ and ln$\dot{\varepsilon }$-σ. According to Equations (6) and (7), the mean slope of the three lines at low stress (880–970 °C) is n_1_ = 8.57, and the mean slope of the three lines at high stress (750–850 °C) is β = 0.127, α can be obtained by β/n_1_, which is 0.015.

Taking the logarithm of both sides of Equation (5) can obtain:(8)$$ln\dot{\varepsilon }=lnA-Q/RT+nln\left[sinh\left(\alpha \sigma \right)\right]$$
by differentiating Equation (8) can obtain:(9)$$Q=R\left\{\frac{\partial \ln\dot{\varepsilon }}{\partial \ln\left[\sinh\left(\alpha \sigma \right)\right]}\right\}{\left\{\frac{\partial \ln\left[\sinh\left(\alpha \sigma \right)\right]}{\partial (1/T)}\right\}}_{\dot{\varepsilon }}$$
where $\left\{\frac{\partial \ln\dot{\varepsilon }}{\partial \ln\left[\sinh\left(\alpha \sigma \right)\right]}\right\}$ is M, which is the slope of the linear relationship between lnsinh(ασ)-ln$\dot{\varepsilon }$; $\left\{\frac{\partial \ln\left[\sinh\left(\alpha \sigma \right)\right]}{\partial (1/T)}\right\}$ is N, which is the slope of the linear relationship between (1/T)-lnsinh(ασ); then Q = R × M × N. The relationship between lnsinh(ασ)-ln$\dot{\varepsilon }$ and (1/T)-lnsinh(ασ) is determined, as shown in Figure 3. The value of Q is 468.5 KJ/mol and n = 7.62, respectively.

Combining Equations (1) and (5), taking the logarithm of both sides and putting Q into it can obtain:(10)lnZ=lnA+nln[sinh(ασ)]

Figure 4 shows the relationship between lnsinh(ασ) and lnZ. A = 2.37 × 10^20^ and n = 7.55 can be obtained from the slope and intercept, respectively, where the value of n is very close to the previous 7.62. In this work, 7.62 is taken as the value of n for subsequent calculations.

It can be concluded that the high-temperature compression of the Cu-1.7Ni-1.4Co-0.65Si alloy is a thermal activation process, and the stress–strain relationship satisfies the constitutive equation of $\dot{\varepsilon }$ = A[sinh(ασ)]^n^exp($\frac{-Q}{\mathrm{RT}}$), which can be expressed as Equation (11):(11)$$\dot{\varepsilon }=e^{46.92}{\left[sinh\left(0.015\sigma \right)\right]}^{7.62}exp(\frac{-468.5}{\mathrm{RT}})$$

### 3.3. The Construction and Analysis of Hot Deformation Processing Maps

The thermal processing maps of the Cu-1.7Ni-1.4Co-0.65Si alloy are constructed under Dynamic Material Modeling (DMM) [17,18,19,20], which is based on the mechanical principle, physical system modeling and irreversible thermodynamics principle of severe plastic deformation. According to the DMM model theory, the energy P consumed by plastic deformation per unit volume of material can be divided into two components, G and J, where G represents the energy consumed by the plastic deformation and J represents the energy consumed by the physical metallurgy mechanism, such as dynamic recovery, dynamic recrystallization and internal defects deformation induced by second phase transformation and precipitation [21,22]. Then, P can be expressed in Equation (12):(12)$$P=G+J=\int_{0}^{\dot{\varepsilon }}\sigma d\dot{\varepsilon }+\int_{0}^{\sigma }\dot{\varepsilon }d\sigma$$

The deformation behavior of alloys is controlled by their internal microstructure and the rate of strain hardening. The flow stress is sensitive to the strain rate and can be expressed in the form of the flow rate:(13)$$\sigma =K^{\dot{\varepsilon }}{}^{m}$$
where *K* is determined by the microstructure and temperature of the deformed alloy; *m* is the strain rate sensitivity coefficient. Under the determined strain and temperature, the value of m can be obtained from Equations (12) and (13):(14)$$m=\frac{dJ}{dG}=\frac{\dot{\varepsilon }d\sigma }{\sigma d\dot{\varepsilon }}={\left|\frac{\partial (\log\sigma )}{\partial (\log\dot{\varepsilon })}\right|}_{\varepsilon ,T}$$

According to the physical meaning of *J* and *G*, the value of m reflects the hot deformation characteristics of alloys. Prasad thought [23,24] that *G* and *J* correspond to the generation and consumption of dislocations, respectively. In the stable flow process, the dislocation consumption rate cannot be greater than the generation rate, so *J* is less than or equal to *G*. According to Equation (12):(15)$$J=\int_{0}^{\sigma }\dot{\varepsilon }d\sigma =\frac{m\sigma \dot{\varepsilon }}{1+m}$$

According to Equation (13), there is a nonlinear relationship between *m*, deformation temperature and strain rate. According to Equation (15), the change of *J* is also nonlinear. The ideal linear dissipation unit, *m* = 1, where the value of *J* is Max, *J_max_*. For the nonlinear dissipation unit, the power dissipation efficiency can be expressed by the ratio of *J* to *J_max_*:(16)$$J_{max}=1/2^{\sigma \dot{\varepsilon }}$$
(17)$$\eta =\frac{J}{J_{max}}=\frac{2m}{1+m}$$

Power dissipation efficiency *η* is a dimensionless parameter and a function of temperature, strain and strain rate. *η* reflects the energy dissipation efficiency of the material.

According to the maximum principle of irreversible thermodynamics, the instability criterion is suitable for large plastic processing. Based on this, Prasad [25] proposed the instability criterion, which can be represented by dimensionless parameter *ξ*($\dot{\varepsilon }$):(18)$$\xi \left(\dot{\varepsilon }\right)=\frac{\partial \mathrm{lg}\left(\frac{m}{1+m}\right)}{\partial lg\dot{\varepsilon }}+m<0$$

The instability diagram is a contour diagram which is composed of the instability criterion, strain rate and deformation temperature. When *ξ*($\dot{\varepsilon }$) < 0, the alloy tends to deform unsteadily, so hot deformation processes should be avoided in this range.

The processing map of the alloy is formed by superposition of the instability diagram and the power dissipation diagram. The processing map is mainly determined by the strain rate sensitivity coefficient m, power dissipation efficiency η and instability criterion parameter *ξ*($\dot{\varepsilon }$), while the power dissipation efficiency η and instability criterion parameter *ξ*($\dot{\varepsilon }$) are all related to the strain rate sensitivity coefficient m, so that the strain rate sensitivity coefficient m is the key to constructing the processing map.

The relationship between logσ and log$\dot{\varepsilon }$ can be obtained as Equation (19):(19)$$log\sigma =a+blog^{\dot{\varepsilon }}+c\;{\left(log^{\dot{\varepsilon }}\right)}^{2}+d{\left(log^{\dot{\varepsilon }}\right)}^{3}$$

Therefore, Equations (13) and (17) can be expressed as:(20)$$m=\frac{d(\log\sigma )}{d(\log\dot{\varepsilon })}=b+2clog^{\dot{\varepsilon }}+3d{\left(log^{\dot{\varepsilon }}\right)}^{2}$$
(21)$$\xi (\dot{\varepsilon })\frac{2c+6d(\log\dot{\varepsilon })}{m(m+1)\ln10}=+m$$

In this paper, the hot deformation behavior of Cu-1.7Ni-1.4Co-0.65Si alloy is analyzed. The deformation temperature and strain rate in different deformation conditions can be put into Equation (20) to obtain the power dissipation diagram and instability diagram, so as to obtain the hot processing map. Figure 5 shows the power dissipation diagram, instability diagram as well as the processing diagram of the alloy when the true strain is 0.8. According to this principle, the processing parameters corresponding to the peak efficiency value in the safe region are the optimal hot processing parameters. It can be concluded from this figure that the reasonable processing parameters of the Cu-1.7Ni-1.4Co-0.65Si alloy are 930–970 °C and 0.01–10 s^−1^, respectively.

### 3.4. Microstructure Analysis

To further analyze the hot compression deformation behavior of the Cu-1.7Ni-1.4Co-0.65Si alloy, the microstructure of different regions is observed. Figure 6 shows the microstructure of the alloy in the instability region. It can be seen that there are some holes in the unstable zone of the alloy at the deformation temperature 760 °C and deformation rate 10 s^−1^, as shown in Figure 6a. This is mainly because the alloy undergoes severe plastic deformation at a high deformation rate and low temperature, hence the rapid deformation ability of alloy is very poor. When the deformation behaviors of adjacent regions are uncoordinated, it is easy to generate large stress concentrations in these uncoordinated regions and form hollow defects. With the deformation temperature increasing, the plastic deformation capacity improves significantly. In addition, the stress in the high strain zone can be released through local deformation, which is conducive to reduce the degree of uncoordinated deformation, thus inhibiting the formation of cracks. As the distortion temperature increases to 910 °C, the storage energy at some locations is very high when there is a high deformation strain rate. Therefore, dynamic recovery and dynamic recrystallization occur at the slip band and grain boundary, as shown in Figure 6b.

The TEM images of a hot specimen at 760 °C with a strain rate of 10 s^−1^ are shown in Figure 7. The plastic deformation of the specimen is very hard when the high strain rate is at a low deformation temperature, and many dislocation cells are observed in the copper matrix from Figure 7a. The pinning effect on the dislocation movement between precipitates and dislocation is significant, as shown in Figure 7b,d, which enhances the strength of the specimen during the deformation process. With the increase in the strain rate, the dislocation entanglement is significant in Figure 7c, which shows the response to the enhancement of flow stress during hot deformation. These precipitates can be confirmed as Ni_2_Si, Co_2_Si or (Ni,Co)_2_Si according to the select area diffraction pattern, as shown in Figure 7b. Based on the literature [26,27,28], the precipitates can be identified as (Ni,Co)_2_Si in the Cu-1.7Ni-1.4Co-0.65Si alloy.

When the hot deformation temperature increases to 910 °C, the density of dislocation clearly decreases and dynamic recrystallization occurs, as shown in Figure 8, which can reduce the work hardening [29,30]. In addition, the fine twins can be found in the matrix according to the select area diffraction pattern from Figure 8c. However, it shows an increase in dislocation density in high deformation strain rate processes, which enhances the interaction between precipitates and dislocations, resulting in an increase in the flow stress [31]. Those are attributed to the serrated feature of the true stress–true strain curve during the hot deformation.

Figure 9 shows the microstructure of the Cu–Ni–Co–Si alloy with a temperature of 940 °C and a strain rate of 10 s^−1^. Compared to the microstructure in the instability zone as shown in Figure 6, there are equiaxial grains and twins in the deformation safety zone. No defects, unstable flow microstructure and holes can be observed and the power dissipation efficiency range is 0.27. The above experimental results show that dynamic recrystallization clearly occurs and fine twin can be found at a high strain rate, as the deformation temperature increases.

## 4. Conclusions

(1)Deformation temperature and strain rate have significant effects on the flow stress of the Cu-1.7Ni-1.4Co-0.65Si alloy. The flow stress increases with the extension of strain rate and decreases with the increase in deformation temperature. The dynamic recovery, recrystallization and work hardening coexist during the hot compression deformation process. The precipitates of the (Ni,Co)_2_Si phase are observed and have a strong effect on dislocation movement, which is a response to the increase in strain and stress.(2)The hot deformation activation energy Q of the Cu-1.7Ni-1.4Co-0.65Si alloy can be calculated as 468.5 kJ/mol, and the high temperature deformation constitutive equation is confirmed as:

(22)$$\dot{\varepsilon }=e^{46.92}{\left[sinh\left(0.015\sigma \right)\right]}^{7.62}exp(\frac{-468.5}{\mathrm{RT}})$$

(3)The hot processing map of the Cu-1.7Ni-1.4Co-0.65Si alloy is established on the basis of hot deformation behavior and hot working characteristics. With optimal hot deformation conditions of 940 to 970 °C and 0.01 to 10 s^−1^, the fine equiaxed grain and no holes are found in the matrix.

## Figures and Tables

**Figure 1 materials-13-02042-f001:**
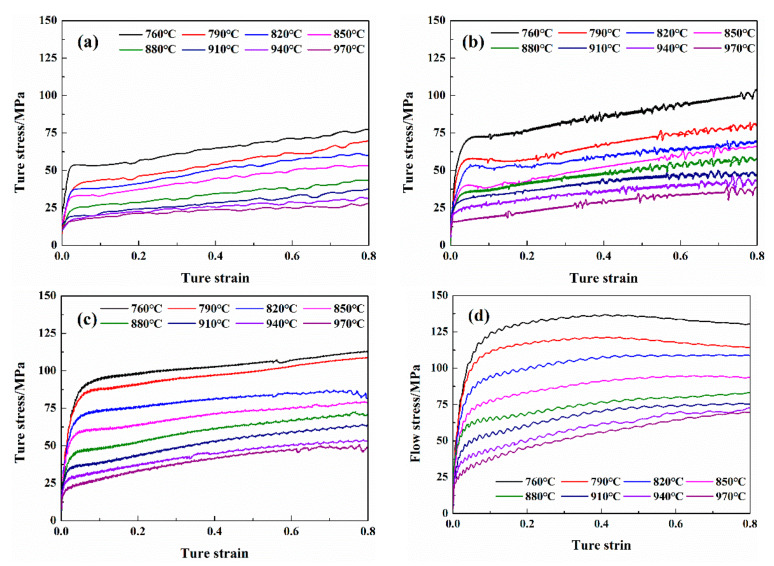
True stress–true strain curves under different deformation temperatures and strain rates: (**a**) $\dot{\varepsilon }$ = 0.01 s^−1^; (**b**) $\dot{\varepsilon }$ = 0.1 s^−1^; (**c**) $\dot{\varepsilon }$ = 1 s^−1^; (**d**) $\dot{\varepsilon }$ = 10 s^−1^.

**Figure 2 materials-13-02042-f002:**
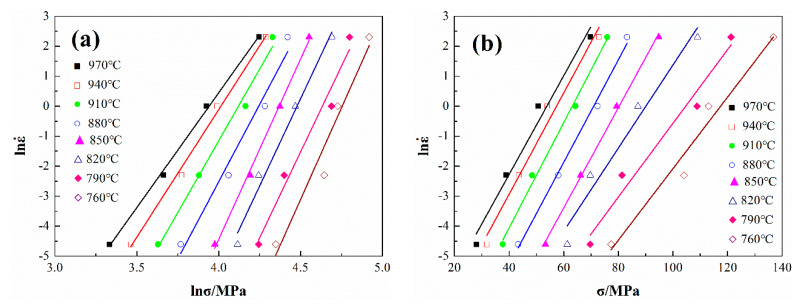
The relationship between ln$\dot{\varepsilon }$-lnσ (**a**) and ln$\dot{\varepsilon }$-σ (**b**).

**Figure 3 materials-13-02042-f003:**
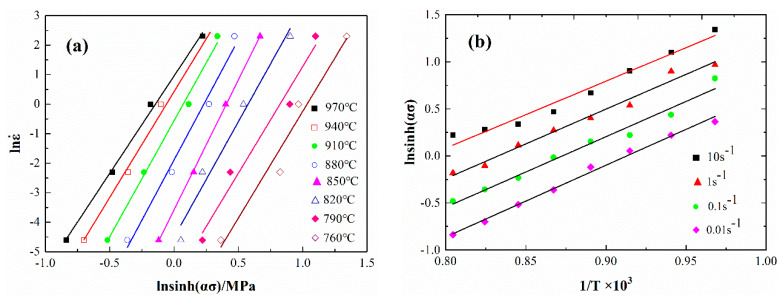
The relationship between lnsinh(ασ)-ln$\dot{\varepsilon }$ (**a**) and (1/T)-lnsinh(ασ) (**b**).

**Figure 4 materials-13-02042-f004:**
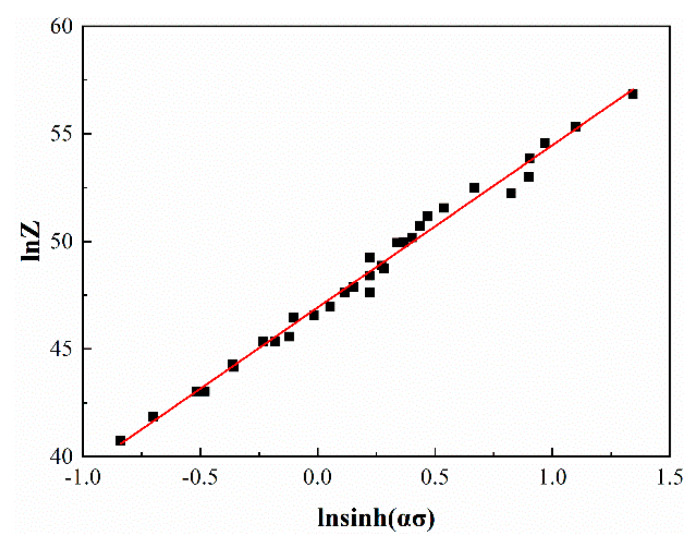
The relationship between lnsinh(ασ) and lnZ.

**Figure 5 materials-13-02042-f005:**
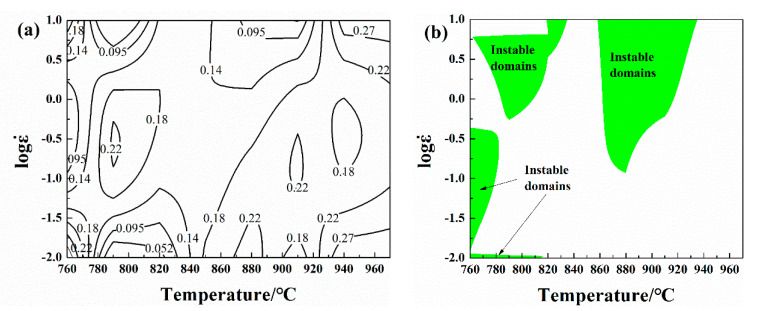
Power dissipation efficiency (**a**) and instable domains (**b**) of the Cu-1.7Ni-1.4Co-0.65Si alloy when the true strain is 0.8.

**Figure 6 materials-13-02042-f006:**
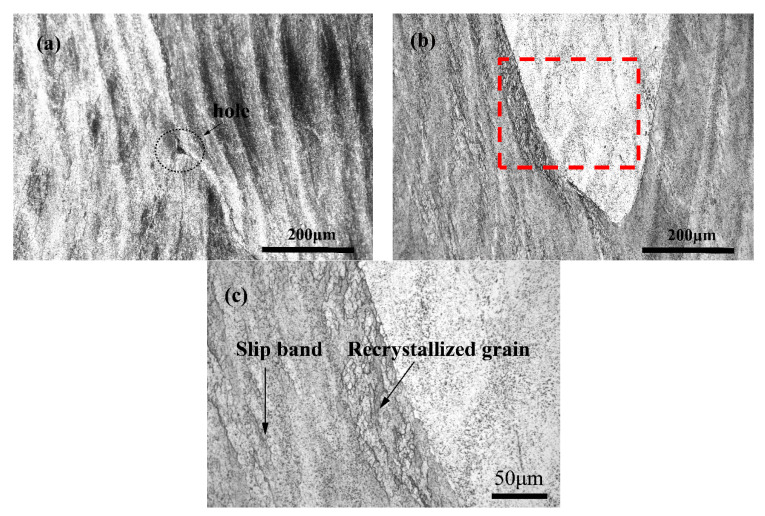
Microstructure of the Cu-1.7Ni-1.4Co-0.65Si alloy at instable regions: (**a**) T = 760 °C, $\dot{\varepsilon }$ = 10 s^−1^; (**b**) T = 910 °C, $\dot{\varepsilon }$ = 10 s^−1^; (**c**) area marked as red rectangle in (b) with magnification 500×.

**Figure 7 materials-13-02042-f007:**
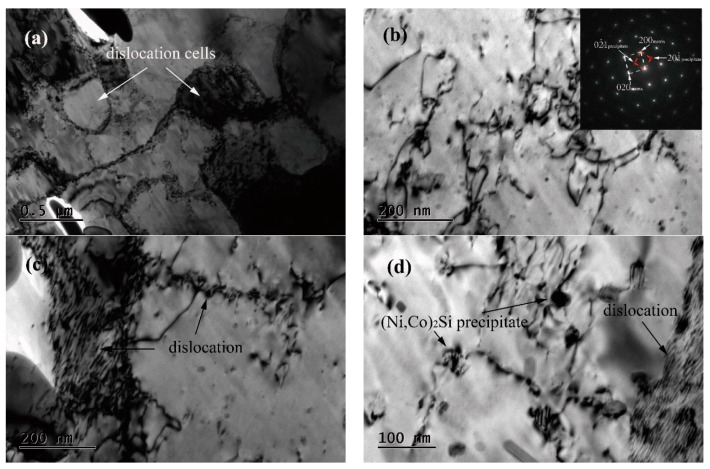
TEM observations of the hot deformed Cu–Ni–Co–Si alloy specimen at a temperature of 760 °C and a strain rate of 10 s^−1^. (**a**,**c**) morphological characteristics of dislocation; (**b**,**d**) relationship between dislocation and precipitation.

**Figure 8 materials-13-02042-f008:**
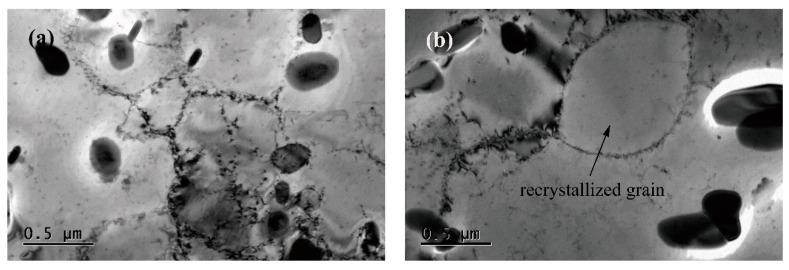
TEM observations of the hot deformed Cu–Ni–Co–Si alloy specimen at a temperature of 910 °C and a strain rate of 10 s^−1^. (**a**,**b**) morphological characteristics of dislocation and recrystallized grain; (**c**) morphological characteristics of Twinning.

**Figure 9 materials-13-02042-f009:**
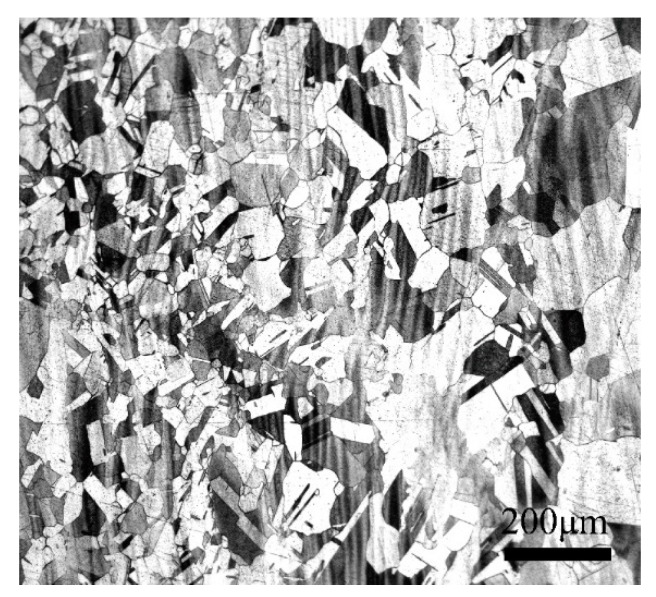
Microstructure observation results of the hot compressed Cu-1.7Ni-1.4Co-0.65Si alloy at a temperature of 940 °C and a strain rate of 10 s^−1^.

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
