# Peer review of "Hot Deformation Behavior and Microstructure Evolution of Cu–Ni–Co–Si Alloys"

_materials, 2020, doi:10.3390/ma13092042_

Round 1

Reviewer 1 Report

Several comments to the authors in order to make the reported results more clear to the readers.

  1. There is no image of the test specimen. It is written that lubrication was applied prior to testing but no image of specimen barreling is shown. If specimen barreling was not significant and could be disregarded in the analysis of flow stress please show this to the readers.
  2. In order to achieve a constant value of strain rate a linear increasing velocity needs to be applied during testing. Was this the case? or are the strain rates reported average strain rate values during the experiments?
  3. It is not clear where in the specimen are the metallurgical images taken from? Since the deformation may be inhomogeneous the position examined must be indicated.   

Author Response

1.There is no image of the test specimen. It is written that lubrication was applied prior to testing but no image of specimen barreling is shown. If specimen barreling was not significant and could be disregarded in the analysis of flow stress please show this to the readers.

The test specimen is a cylindera with a diameter of 10 mm and a length of 15 mm. The specimen barreling is not significant during the hot deformation.

2.In order to achieve a constant value of strain rate a linear increasing velocity needs to be applied during testing. Was this the case? or are the strain rates reported average strain rate values during the experiments?

The strain rates are average strain rate values.

3.It is not clear where in the specimen are the metallurgical images taken from? Since the deformation may be inhomogeneous the position examined must be indicated.   

The metallurgical images are taken from the center of specimen.

Reviewer 2 Report

The aim of the paper, i.e. the study of hot deformation behavior of Cu-1.7 wt% Ni-1.4 wt% Co-0.65 wt% Si alloy, is quite interesting. Moreover, authors established the constitutive relationship of high temperature deformation and hot processing map to reveal the microstructure evolution and deformation mechanism during the hot deformation process, and to provide a guidance for the formulation of the hot processing technology of Cu-Ni-Co-Si alloy.

The abstract summarize the work. The purpose of the study is clearly outlined and the findings of prior work are well discussed. There are no major errors in logic or experimental procedure. The authors accurately explain how the data were collected. There is sufficient information that the experiment can be reproduced. All topics are well presented and discussed. The summary and conclusions are sound and justified. All presented figures are good quality and they prove their point. The paper is written in good English. The manuscript is easily readable concerning language, style and presentation. The references are appropriate and up to date.

Author Response

Some English language has been modified in the paper. 

Reviewer 3 Report

This article deals with Cu-Ni-Co-Si alloy and reports microstructure evolution of hot deformed samples. Authors developed constitutive model of flow stress and performed hot deformation processing maps for mentioned above alloy. However analysis of microstructure evolution should be conducted in different way. There is the lack of SEM observation and EBSD measurement (grain morphology, texture, etc.). TEM observation/analysis should be used as detailing of SEM/EBSD results. Moreover in the article titled “Effects of Cr addition on the constitutive equation and precipitated phases of copper alloy during hot deformation” authored by Yijie Ban et al. (Materials & Design, Volume 191, June 2020, 108613) similar research has been presented. Due to Authors must be clearly indicate article novelty and carefully check it for plagiarism.

Author Response

  1. In page 1/11 in the introduction, before “Lei has stadied… “ Add the following statement using the related reference;“ The effect of hot deformation on microstructure and mechanical properties of copper alloy was studied in the previous studies.”

The following statement can be added in the paper

  1. In page 1/11 in the introduction, before “Since the phase transition temperature… “, It is suggested to introduce the formation condition and properties of (Ni,Co)2Si and Ni2Si before direct discussion about their effect on deformation behavior of alloys.

The Ni2Si phase can be observed in the Cu-Ni-Si alloy. The addition of Co to the Cu-Ni-Si alloy, (Ni,Co)2Si phase can be also observed in the Cu-Ni-Co-Si alloy, because the combination capacity between Ni or Co and Si is very close. Since the phase transition temperature and stress of (Ni, Co)2Si phases are higher than that of Ni2Si phase. The high temptation deformation of Cu-Ni-Co-Si alloy is much higher than Cu-Ni-Si alloy.

  1. In page 2/11 in the results and discussion, ”When the stress reaches the peak…” But the peaks of the curves are not shown in the related Figure 1 (a-d). Could author show the full scale of curves until fracture to understand the UTS and elongation of alloys.

The full stress curve was obtained when the strain rate is set as 0.8. The full scale of curves until fracture are not obtained. Some peaks of the curves are found in the Figure.1(d) and these curves show same change rule between stress and strain. Although some peaks of curves are not shown in Figure 1, the change rule of those curves can be inferred according to this work or previous research work.

  1. In page 2/11 in the results and discussion, in last paragraph author discuss about the serrated behavior of S-S curve in Fig. d. The fluctuations was explained by softening and hardening during deformation. Can you explain more about the reason and/or show the similar behavor in the previous research works? Also, It seems there is a typo error in Fig.1. Both axes show the stress values. X axis should be Strain?

According to this work and previous research works, the fluctuations were mainly caused by softening and hardening during deformation. The Fig.1 had been modified in the paper.

  1. In page 9/11 in the results and discussion, there is a discrepancy in the second paragraph, In the first sentence reduced dislocation density was mentioned when explaining Fig. 8 (a-b) but later in the third sentence increasing of dislocation density was the casue of stress increasing in the same Fig. On the other hand, due to low strength of alloy processed at 910 C, you can eliminate the following sentence “However, the ointeraction between…”

The dynamic recrystallization and working hardening were observed during the hot deformation. The former can reduce dislocation density to decrease the stress and the later can increase the dislocation density to increase the stress. The stress of curve can be determined according to microstructure evolution between dynamic recrystallization and working hardening

  1. Also can you confirm the presence of twining in Fig. 8(c)? it is not like twining, or you can remove it.

The twining can be confirmed in Fig.8(c) according to the select area diffraction pattern of typical twining.

  1. Dislocation density and pining effect is not the only strengthening mechanism of deformed alloys, but also you should considerthe effect of grain size based on Hall-Petch effect. Therefore, it is suggested to add the microstructure image of alloy processed at 760 C and 910 C to Fig. 9 and discuss the effect of grain size using the following reference. Additionally, why you show the microstructure of alloy at 940 C while you have compared and disccussed the microstructure and strength of alloys at 760 C and 910 C in all discussions? Therefore, it seems showing the microstructure of 940 C is not consistent with other discussed parts of manuscript.

The recrystallization process can be not obviously, when the alloy processed at 760 ℃ and 910℃ from the Fig.6. The average grain size can not be accurately calculated and the effect of grain size on properties can not be discussed based on Hall-Petch. The microstructure of 940 ℃ to show the reasonable microstructure when the temperature located in the safety zoon, which show a good guidance to determination of hot deformation system.

  1. Add more about the effect of microstructure in the conclusion section.

The effect of microstructure in the conclusion section can be added in the paper.

Reviewer 4 Report

The authors presented an experimental research on hot deformation behavior and microstructure evolution of Cu-Ni-Co-Si alloya. However, there are still some comments need to be addressed.

1- In page 1/11 in the introduction, before “Lei has stadied… “ Add the following statement using the related reference;

“ The effect of hot deformation on microstructure and mechanical properties of copper alloy was studied in the previous studies.”

[Ref] Synergistic strengthening mechanisms of copper matrix composites with TiO2 nanoparticles.

2- In page 1/11 in the introduction, before “Since the phase transition temperature… “, It is suggested to introduce the formation condition and properties of (Ni,Co)2Si and Ni2Si before direct discussion about their effect on deformation behavior of alloys.

3- In page 2/11 in the results and discussion, ”When the stress reaches the peak…” But the peaks of the curves are not shown in the related Figure 1 (a-d). Could author show the full scale of curves until fracture to understand the UTS and elongation of alloys.

4- In page 2/11 in the results and discussion, in last paragraph author discuss about the serrated behavior of S-S curve in Fig. d. The fluctuations was explained by softening and hardening during deformation. Can you explain more about the reason and/or show the similar behavor in the previous research works? Also, It seems there is a typo error in Fig.1. Both axes show the stress values. X axis should be Strain?

5- In page 9/11 in the results and discussion, there is a discrepancy in the second paragraph, In the first sentence reduced dislocation density was mentioned when explaining Fig. 8 (a-b) but later in the third sentence increasing of dislocation density was the casue of stress increasing in the same Fig. On the other hand, due to low strength of alloy processed at 910 C, you can eliminate the following sentence “However, the ointeraction between…”

Also can you confirm the presence of twining in Fig. 8(c)? it is not like twining, or you can remove it.

6- Dislocation density and pining effect is not the only strengthening mechanism of deformed alloys, but also you should considerthe effect of grain size based on Hall-Petch effect. Therefore, it is suggested to add the microstructure image of alloy processed at 760 C and 910 C to Fig. 9 and discuss the effect of grain size using the following reference. Additionally, why you show the microstructure of alloy at 940 C while you have compared and disccussed the microstructure and strength of alloys at 760 C and 910 C in all discussions? Therefore, it seems showing the microstructure of 940 C is not consistent with other discussed parts of manuscript.

[Ref] Asymmetric local strain, microstructure and superelasticity of friction stir welded Nitinol alloy

7- Add more about the effect of microstructure in the conclusion section.

Author Response

(The authors gave the same response as above.)

Round 2

Reviewer 3 Report

Dear Authros,

Please apply to my comments:

"This article deals with Cu-Ni-Co-Si alloy and reports microstructure evolution of hot deformed samples. Authors developed constitutive model of flow stress and performed hot deformation processing maps for mentioned above alloy. However analysis of microstructure evolution should be conducted in different way. There is the lack of SEM observation and EBSD measurement (grain morphology, texture, etc.). TEM observation/analysis should be used as detailing of SEM/EBSD results. Moreover in the article titled “Effects of Cr addition on the constitutive equation and precipitated phases of copper alloy during hot deformation” authored by Yijie Ban et al. (Materials & Design, Volume 191, June 2020, 108613) similar research has been presented. Due to Authors must be clearly indicate article novelty and carefully check it for plagiarism."

Kind regards,

Rewiever

Author Response

In this paper, the Cu-Ni-Co-Si alloy ingot was prepared by using an intermediate frequency induction melting furnace with the size of 210 mm (width) × 640 mm (length) × 6000 mm (height). The preparation method of the former is different from the article titled “Effects of Cr addition on the constitutive equation and precipitated phases of copper alloy during hot deformation”. Different preparation methods of copper alloy have different microstructure. The microstructure evolution of Cu-Ni-Co-Si alloy during the hot deformation at temperature 760 to 970 ℃ and strain rate range 0.01 to 10s-1 are studied in detail in this work. However, the effect of Cr addition on microstructure during the hot deformation at temperature 500 to 900 ℃ and strain rate range 0.001 to 1s-1 are discussed in Yijie Ban’s article. The results show that the fine equiaxed grain and no holes were found in the matrix at the optimal hot deformation condition of 940 to 970 °C and 0.01 to 10 s–1, which provides a guidance for the formulation of the hot processing technology of Cu-Ni-Co-Si alloy in the industrialization production. The latter is focused on the effect of Cr on the microstructure evolution during the hot deformation.

In addition, the microstructure evolution during the hot deformation are investigated in detail by OM and TEM in this paper. The microstructure characterization can be clearly observed and characterized. The analysis of microstructure evolution will be conducted by EBSD and SEM in the future, if it is necessary.

Reviewer 4 Report

Some comments has been replied properly, but the following are still need to be addressed accordingly. 

1-In page 1/11 in the introduction, before “Lei has studied… “

Add the following statement using the related reference;

“The effect of hot deformation on microstructure and mechanical properties of copper alloy was studied in the previous studies.”

[Ref] Synergistic strengthening mechanisms of copper matrix composites with TiO2 nanoparticles

The following statement can be added in the paper

This comment still not included in the manuscript. It is suggested to improve the introduction. Please follow the comment and ref.

2-In page 1/11 in the introduction, before “Since the phase transition temperature… “, It is suggested to introduce the formation condition and properties of (Ni,Co)2Si and Ni2Si before direct discussion about their effect on deformation behavior and etc. of alloys.

The Ni2Si phase can be observed in the Cu-Ni-Si alloy. The addition of Co to the Cu-Ni-Si alloy, (Ni,Co)2Si phase can be also observed in the Cu-Ni-Co-Si alloy, because the combination capacity between Ni or Co and Si is very close. Since the phase transition temperature and stress of (Ni, Co)2Si phases are higher than that of Ni2Si phase. The high temptation deformation of Cu-Ni-Co-Si alloy is much higher than Cu-Ni-Si alloy.

The comment has been replied but was not included in the manuscript! Please modify and add it to the manuscript as explained in the initial review comment. You can improve slightly your above statement as follows:” According to the trinary phase diagram of Cu-Ni-Si, Ni2Si intermetallic can be precipitates at temperature….the addition of Co to this system also results in formation of (Ni,Co)2Si. Since…”

3- In page 2/11 in the results and discussion, ”When the stress reaches the peak…” But the peaks of the curves are not shown in the related Figure 1 (a-d). Could author show the full scale of curves until fracture to understand the UTS and elongation of alloys.

The full stress curve was obtained when the strain rate is set as 0.8. The full scale of curves until fracture are not obtained. Some peaks of the curves are found in the Figure.1(d) and these curves show same change rule between stress and strain. Although some peaks of curves are not shown in Figure 1, the change rule of those curves can be inferred according to this work or previous research work.

The trend of curves are increasing in most of the curves (except some in Figure (d) reaching 0.8 strain and can not confirm that stress level stay stable at this point and do not goes up! However, to help reader understanding it is suggested to add your first sentence of response in the manuscript before discuss about curves. “The full stress curve was obtained when the strain rate is set as 0.8”.

4- Dislocation density and pining effect is not the only strengthening mechanism of deformed alloys, but also you should consider the effect of grain size based on Hall-Petch effect. Therefore, it is suggested to add the microstructure image of alloy processed at 760 C and 910 C to Fig. 9 and discuss the effect of grain size using the following reference. Additionally, why you show the microstructure of alloy at 940 C while you have compared and disccussed the microstructure and strength of alloys at 760 C and 910 C in all discussions? Therefore, it seems showing the microstructure of 940 C is not consistent with other discussed parts of manuscript.

[Ref] Asymmetric local strain, microstructure and superelasticity of friction stir welded Nitinol alloy

The recrystallization process can be not obviously, when the alloy processed at 760 ℃ and 910℃ from the Fig.6. The average grain size can not be accurately calculated and the effect of grain size on properties can not be discussed based on Hall-Petch. The microstructure of 940 ℃ to show the reasonable microstructure when the temperature located in the safety zoon, which show a good guidance to determination of hot deformation system.

First, if there is no obvious dynamic recrystallization at 760 ℃ and 910℃ but still similar behavior of softening and hardening can be seen in their S-S curves indicating recrystallization and work hardening as you confirmed. On the other hand, recrystallization is shown in the Fig. 8 (b).  Considering the title of this article on microstruction evolution of hot deformed Cu-Ni-Si ally and by understanding that hot deformation normally triggers grain refinement compare to the reference undeformed Cu-Ni-Si ally. Therefore, it is suggested to compare the microstructure of 760 ℃ or 910℃ with undeformed Cu-Ni-Si ally briefly and no need to measure the grain size and just address grains are smaller after hot deformation. (However, you can measure the grain size using intercept method according to ASTM E-112 if you want).

This discussion will support your result on higher strength in S-S curve of particularly 760 ℃.

For example, you can add this way ”hot deformed 760 ℃ showed smaller grains compare to the reference Cu-Ni-Si ally. This can contribute to improve the strength after hot deformation according to Hall-Petch relationship as follows; ADD the formula here”.

5- Also pg 2/12 consider to revise "state stage" in the following:

At this state stage, the deformation is in a relative equilibrium state with the combination of 85 dynamic softening and work hardening.

Author Response

1.This comment still not included in the manuscript. It is suggested to improve the introduction. Please follow the comment and ref.

The comment and ref is added in the paper.

2.The comment has been replied but was not included in the manuscript! Please modify and add it to the manuscript as explained in the initial review comment. You can improve slightly your above statement as follows:” According to the trinary phase diagram of Cu-Ni-Si, Ni2Si intermetallic can be precipitates at temperature….the addition of Co to this system also results in formation of (Ni,Co)2Si. Since…”

The comment is added in the paper

3.The trend of curves are increasing in most of the curves (except some in Figure (d) reaching 0.8 strain and can not confirm that stress level stay stable at this point and do not goes up! However, to help reader understanding it is suggested to add your first sentence of response in the manuscript before discuss about curves. “The full stress curve was obtained when the strain rate is set as 0.8”.

The sentence is added in the paper.

4.

This discussion will support your result on higher strength in S-S curve of particularly 760 ℃.

For example, you can add this way ”hot deformed 760 ℃ showed smaller grains compare to the reference Cu-Ni-Si ally. This can contribute to improve the strength after hot deformation according to Hall-Petch relationship as follows; ADD the formula here”.

The related discussion is added in the paper.

5.At this state stage, the deformation is in a relative equilibrium state with the combination of 85 dynamic softening and work hardening.

This sentence can be deleted in the paper

Round 3

Reviewer 3 Report

Authors carefully explained all doubts. However SEM observations and EBSD measurements would more precisely show evaluation of samples microstructure after deformation at different temperature. In my opinion it is worth to do in the nearest future. Summarizing, the article can be publishing after correction few minor/major mistakes, i.e.:

  1. Line 76 – Please use Light Microscope instead of Optical Microscope – check throughout the article
  2. Figure 1 – The caption of Y-axis and X-axis should be True stress and True strain/ε (the X-axis presents true strain [ε] at different strain rate [ε. ]) respectively. The caption of Figure 1– It should be True stress.
  3. Figure 6 – the quality must be strongly improved, the descriptions/selected areas are not visible enough. Please add the unit for strain rate (see figure caption).
  4. Figure 7 - “TEM observations of hot Cu-Ni-Co-Si alloy specimen…” should be “TEM observations of hot deformed Cu-Ni-Co-Si alloy specimen…”
  5. Figure 8 – TEM pattern should be solved (Fig.8c). “TEM observations of hot Cu-Ni-Co-Si alloy specimen…” should be “TEM observations of hot deformed Cu-Ni-Co-Si alloy specimen…”
  6. Figure 9 – Why is the grains size non-homogeneity? Where is the top of the sample?
  7. Line 255, 256 – What does “harden working” mean??
  8. Line 263 – “…the fine equiaxed grain and no holes were found in the matrix.” Please explain - What “holes” do you mean? What is about non-homogeneity of grains size?
  9. Line 263-264 – “With the optimal hot deformation condition of 940 to 970 °C and 0.01 to 10 s–1…” – only the variants 760oC, 10s-1; 910oC, 10s-1 and 940oC, 10s-1 have been showed. The variant at 970oC, 10s-1 has not been described anywhere.

Author Response

1. Line 76 – Please use Light Microscope instead of Optical Microscope – check throughout the article Optical Microscope is a fixed collocation, which should not be changed. 2. Figure 1 – The caption of Y-axis and X-axis should be True stress and True strain/ε (the X-axis presents true strain [ε] at different strain rate [ε. ]) respectively. The caption of Figure 1– It should be True stress. The Figure 1 has been advised in the paper. 3. Figure 6 – the quality must be strongly improved, the descriptions/selected areas are not visible enough. Please add the unit for strain rate (see figure caption). The Figure caption has been added in the paper. 4. Figure 7 - “TEM observations of hot Cu-Ni-Co-Si alloy specimen…” should be “TEM observations of hot deformed Cu-Ni-Co-Si alloy specimen…” It has been advised in the paper 5. Figure 8 – TEM pattern should be solved (Fig.8c). “TEM observations of hot Cu-Ni-Co-Si alloy specimen…” should be “TEM observations of hot deformed Cu-Ni-Co-Si alloy specimen…” It has been advised in the paper 6. Figure 9 – Why is the grains size non-homogeneity? Where is the top of the sample? The microstructure of the middle of the hot deformation sample are observed. From the Figure.9, some areas of grains size is non-homogeneity, but that of whole sample are homogeneity.  7. Line 255, 256 – What does “harden working” mean?? It is wrong. It has been advised as Working hardening 8. Line 263 – “…the fine equiaxed grain and no holes were found in the matrix.” Please explain - What “holes” do you mean? What is about non-homogeneity of grains size? Holes stand for defects caused during the hot deformation. The grains size of whole sample is homogeneity. 9. Line 263-264 – “With the optimal hot deformation condition of 940 to 970 °C and 0.01 to 10 s–1…” – only the variants 760oC, 10s-1; 910oC, 10s-1 and 940oC, 10s-1 have been showed. The variant at 970oC, 10s-1 has not been described anywhere. Because there are equiaxial grains and twins in the deformation safety zone from the sample which is hot deformed at temperature of 970 ℃ and strain rate of 10 s-1. The optimal thermal deformation conditions can be confirmed. From the sample deformed at temperature of 970 ℃ and strain rate of 10 s-1, the equiaxial grains and no defects can be observed. 1.  Microstructure observation results of hot compressed Cu-1.7Ni-1.4Co-0.65Si alloy at temperature of 970 ℃ and strain rate of 10 s-1.

Round 4

Reviewer 3 Report

Dear Authors,

Some points of my recommendations have been omitted:

  1. Figure 1 – The labels of Y-axis and X-axis must be True stress and True strain respectively. Moreover True strain has not unit. Please leave only True strain label. Please check the figure caption – should be “True stress-true strain…” – check it carefully.
  2. Figure 6 – Please add the unit for strain rate in figure caption. Moreover caption for c) sounds strange - “area marked as red rectangle in b) with magnification …x” sounds more better.
  3. Figure 8c – TEM pattern is still unsolved. Please correct it.
  4. Line 239 – Should be “work hardening” instead of “hardening work”.

Author Response

  1. Figure 1 – The labels of Y-axis and X-axis must be True stress and True strain respectively. Moreover True strain has not unit. Please leave only True strain label. Please check the figure caption – should be “True stress-true strain…” – check it carefully.

The Fig.1 has been modified in the paper.

  1. Figure 6 – Please add the unit for strain rate in figure caption. Moreover caption for c) sounds strange - “area marked as red rectangle in b) with magnification …x” sounds more better.

The Fig.6 has been modified in the paper.

  1. Figure 8c – TEM pattern is still unsolved. Please correct it.

TEM pattern has been solved in the paper.

  1. Line 239 – Should be “work hardening” instead of “hardening work”.

It has been modified in the paper